# Direct calibration using atmospheric particles and performance evaluation of PSM 2.0 for sub-10 nm particle measurements

Yiliang Liu[1,2], Arttu Yli-Kujala[2], Fabian Schmidt-Ott[2], Sebastian Holm[2], Lauri Ahonen[2], Tommy Chan[2], Joonas Enroth[3], Joonas Vanhanen[3], Runlong Cai[4], Tuukka Petäjä[2], Markku Kulmala[2], Yang Chen[1], and Juha Kangasluoma[2]

[1]Chongqing Institute of Green and Intelligent Technology, CAS, Chongqing, 400714, China
[2]Institute for Atmospheric and Earth System Research / Physics, University of Helsinki, Helsinki, 00014, Finland
[3]Airmodus Ltd., Helsinki, 00560, Finland
[4]Department of Environmental Science & Engineering, Fudan University, Shanghai, 200438, China

*Correspondence to*: Juha Kangasluoma (juha.kangasluoma@helsinki.fi), and Yang Chen (chenyang@cigit.ac.cn)

**Abstract.** Particle Size Magnifier is widely used for the measuring nano-sized particles. Here we calibrated the newly developed Particle Size Magnifier version 2.0 (PSM 2.0). 1-10 nm particles with different compositions were used, including metal particles, organic particles generated in the laboratory and atmospheric particles collected in Helsinki and Hyytiälä, respectively. Noticeable difference among the calibration curves was observed. Atmospheric particles from Hyytiälä required higher DEG supersaturation to be activated compared to metal particles (standard calibration particles) and other types of particles. This suggests that chemical composition differences introduce measurement uncertainties and highlight the importance of in-situ calibration. The size resolution of PSM 2.0 was characterized using metal particles. The maximum size resolution was observed at 2-3 nm. PSM 2.0 was then operated in Hyytiälä for ambient particle measurements in parallel with a Half-mini Differential Mobility Particle Sizer (DMPS). During new particle formation (NPF) events, comparable total particle concentrations were observed between Half-mini DMPS and PSM 2.0 based on Hyytiälä atmospheric particle calibration. Meanwhile, applying the calibration with metal particles to atmospheric measurements would cause an overestimation of 3-10 nm particles. In terms of the particle size distributions, similar patterns were observed between DMPS and PSM when using the calibration of Hyytiälä atmospheric particles. In summary, PSM 2.0 is a powerful instrument for measuring sub-10 nm particles and can achieve more precise particle size distribution measurements with proper calibration.

## 1 Introduction

Particle Size Magnifier (PSM) is a powerful instrument for measuring the size distributions of nanoparticles. It finds extensive applications across various research domains, such as atmospheric studies (Kulmala et al., 2013; Winkler et al., 2008; Yao et al., 2018), nano material research (Liu et al., 2024; Wlasits et al., 2020), combustion research (Rönkkö et al., 2017), health sciences, etc. The prototype instrument was developed by Kogan et al. in 1960 (Kogan and Burnasheva, 1960), followed by a series of different types of designs (Kim et al., 2003; Okuyama et al., 1984; Sgro and Fernández de la Mora, 2004). In 2011,

PSM version 1.0 was commercialized by Airmodus Ltd. (Vanhanen et al., 2011), and became available for research groups worldwide. In 2023, Airmodus Ltd. unveiled PSM version 2.0, boasting several enhancements over its predecessor (Sulo et al., 2024). These improvements include an expanded measurement size range, enhanced durability, higher instrument stability, and improved user-friendliness.

The activation of particles in PSM relies on heterogeneous nucleation under varying supersaturation of diethylene glycol (DEG) vapor. As DEG supersaturation increases, smaller particles become activated. The minimum particle size that can be activated is around 1 nm, at the brink of DEG homogeneous nucleation (Iida et al., 2011). However, the performance of PSM, as well as some other condensation-based instruments, is influenced by many factors. A review paper that summarizes and discusses the uncertainties is published recently (Kangasluoma et al., 2020). The uncertainties in particle activation come from the particle properties, ambient conditions, and instrument setups, etc. Environmental conditions such as humidity and air pressure can result in the offset between the DEG supersaturation and particle activation, and thereby introduce uncertainties for the measured particle size distributions (Liu et al., 2020). As the relative humidity (RH) of the carrier gas increases, nanoparticles can be activated at a lower DEG saturator flow rate, leading to a shift in the cut-off size towards smaller particles (Kangasluoma et al., 2013, 2016b). This phenomenon is attributed to the hygroscopic properties of nanoparticles and vapor-particle interactions due to hydrogen bonding (Keshavarz et al., 2020b; Toropainen et al., 2021). The charging state of particles also affects their activation. Charged particles tend to activate at a lower saturator flow rate compared to neutral particles (Kangasluoma et al., 2016a; Keshavarz et al., 2020a), because the charge on a particle reduces the energy barrier for DEG condensation.

Among all the influencing factors, the chemical composition of particles exerts the strongest influence on particle detection (Kulmala et al., 2007). Metal and salt particles typically exhibit higher detection efficiency by PSM compared to organic particles (Kangasluoma et al., 2014; Liu et al., 2022). The standard practice involves calibrating the PSM in a laboratory using metal particles and then employing it for measuring particles with complex and unknown compositions (Lehtipalo et al., 2021). This process inherently introduces some uncertainties due to the variation in chemical composition of particles used for calibration and measurement. Calibrating the PSM with particles of the same chemical composition as those being measured can minimize measurement uncertainties (Ahonen et al., 2017). However, applying this approach can be challenging, as it requires specific and directly relevant particle sources and a HR-DMA, which are not always readily available. Additionally, the measured nanoparticles often have complex compositions and low concentrations, making it difficult to generate particles with identical compositions in the laboratory.

In this study, we performed calibration on PSM 2.0 using different kinds of particles, including commonly-used metal particles, organic particles, as well as the direct calibration using atmospheric particles. In addition, PSM 2.0 was operated in parallel with a DMPS (Differential Mobility Particle Sizer) for the measurement of ambient particles. The results help us to have a better understanding about the effects of particle composition as well as the sizing accuracy of PSM 2.0.

## 2 Methods

### 2.1 Working principle of PSM 2.0

The working principle of PSM 2.0 is similar to that of its predecessor, PSM 1.0, involving a two-step activation and growth process for nanoparticles. First, particles are activated using supersaturated DEG vapor, where increased DEG supersaturation enables the activation of smaller particles. In the second step, the activated particles grow to an optically detectable size within a butanol-based CPC. The size distributions can be calculated through a proper data inversion process, and in this study the step inversion method was used (Cai et al., 2018b; Chan et al., 2020).

PSM 2.0 offers several improvements compared over PSM 1.0. Firstly, it features a more stable internal flow field, achieved by controlling DEG supersaturation through two distinct flows: a wet flow carrying DEG vapors and a dry flow of particle-free compressed air (Attoui et al., 2023). The wet flow rate can be adjusted between 0.05 and 1.90 L min$^{-1}$, while maintaining a stable total flow rate of wet and dry flows. The enhanced stability and precision in the flow system lead to more predictable dilution factors under different saturator flow rates and minimize concentration fluctuations caused by changing flow rates. Secondly, PSM 2.0 offers a broader range of DEG supersaturation adjustments than PSM 1.0. By controlling the ratio of dry to wet flow rates, it enables to achieve a lower supersaturation level, which is crucial for extending the instrument's size measurement range.

### 2.2 PSM 2.0 calibration

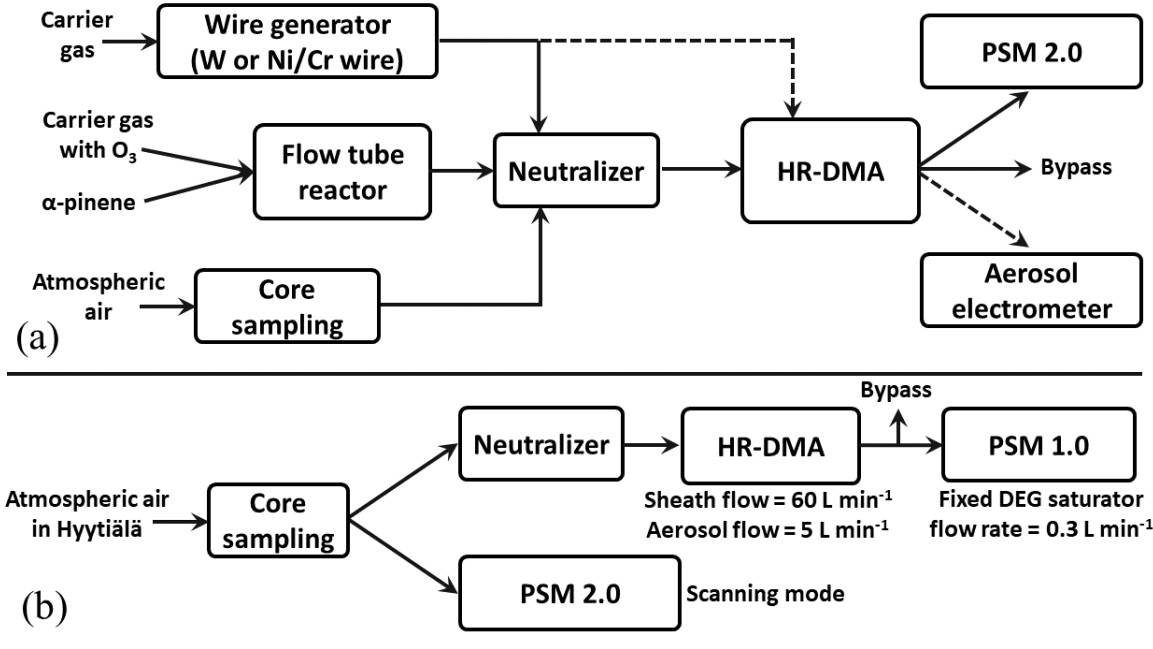

**Figure 1. (a) A schematic diagram illustrating the calibration of PSM 2.0 using metal particles (tungsten and nickel/chromium particles), organic particles (alpha-pinene oxidation particles), and atmospheric particles collected in Helsinki and Hyytiälä. The flow configuration is detailed in Table 1. (b) PSM 2.0 operated in parallel with a DMPS to measure atmospheric particles in Hyytiälä. The HR-DMA operated with an aerosol flow rate of 5 L min⁻¹ and a sheath flow rate of 60 L min⁻¹.**

A schematic diagram for PSM 2.0 calibration is displayed in Fig. 1(a). Different types of particles were used, including tungsten (W) metal particles, nickel/chromium (Ni/Cr) particles, alpha-pinene oxidation particles, Helsinki atmospheric particles and Hyytiälä atmospheric particles (Table 1). Metal particles generated in wire generators are commonly used for PSM calibration, because they provide appropriate concentrations and their activation behaviour is similar to that of salt particles, which are assumed to be one of the major components of ambient aerosols (Kangasluoma et al., 2014, 2015). For the tungsten particle

generation, nitrogen was used as the carrier gas, while synthesized air was used for Ni/Cr particle generation. Tungsten particles ranged in size from 1.2 to 20.3 nm across 68 sizes, whereas Ni/Cr particles ranged from 1.2 to 14.0 nm across 10 sizes. Organic particles were generated in the flow tube reactor through the reaction of alpha-pinene with ozone. The generation method was comparable to that mentioned in a previous study (Li et al., 2022). The particles, ranging in size from 1.6 to 9.4 nm, were classified into 10 sizes and used for PSM 2.0 calibration. The compositional differences between laboratory-generated particles

and atmospheric particles have been the main source of uncertainty in PSM measurements. Ambient particle compositions vary significantly across different locations, highlighting the value of performing direct calibration using atmospheric particles. A burst increase in sub-10 nm particle concentrations was observed during NPF events in Helsinki and Hyytiälä, with atmospheric particles being sampled directly for the calibration of PSM 2.0. After the particle generation, the aerosol flow passed through a neutralizer (Ni-63) before undergoing size classification by an HR-DMA (Cai et al., 2018a; Fernández de la

Mora, 2017). Some of the metal particles were self-charged, so the neutralizer was removed when using sub-3 nm tungsten to calibrate the instrument. It will help to eliminate the effects of neutralizer ions of the calibration. High sheath flow rates were generally used for classifying metal particles to enhance their monodispersity in the laboratory experiments (Table 1). A decreased DMA sheath/aerosol flow rate ratio of 60/5 L min⁻¹/ L min⁻¹ was used for the classification of atmospheric particles, which would help to increase the penetration efficiency of nanoparticles passed through the DMA, and obtain a sufficient

concentration for calibration.

    After the HR-DMA classification, the classified particles were directed to PSM 2.0 and an Aerosol Electrometer (AEM) at equal flow rates of 2.7 L min⁻¹. PSM 2.0 was operated in scanning mode, with the DEG saturator flow rate(s) repeatedly increasing and decreasing between 0.05 and the peak value (usually 1.9 L min⁻¹). Particle concentrations at different DEG saturator flow rates were recorded. The AEM was only used for the calibration using metal particles and organic particles,

when particle concentrations exceeded 500 cm⁻³. For the calibration using atmospheric particles, the particle concentrations after the DMA classification were very low (<100 cm⁻³). The concentrations were lower than the background noise concentrations from the AEM. In this case, the identification of true particle concentrations relies on the concentrations measured at different DEG saturator flow rates. For particles larger than 2 nm, plateau concentrations measured at high DEG saturator flow rates represented the true particle concentrations.

The standard temperature settings by the manufacturer were used for all calibration experiments, and one more calibration was performed under boosted temperature setting in Hyytiälä. The temperatures for both the standard and boosted settings are displayed in Table 2. The standard temperature setting established a condenser temperature of 10°C to reduce the potential co-condensation between diethylene glycol (DEG) and water molecules at peak DEG saturator flow rates. The saturator temperature was carefully adjusted to ensure that the scanning DEG saturator flow rate (from 0.05 to 1.90 L min$^{-1}$) could activate particles both near 1 nm and larger than 10 nm. The boosted temperature setting approached the threshold for DEG homogeneous nucleation at peak DEG saturator flow rates, enhancing PSM's detection efficiency for particles closing to 1 nm, meanwhile reducing the upper size limit.

**Table 1. Summary of particles used for PSM 2.0 calibration. The particles were either generated in the laboratory using particle generators or sampled from atmosphere during NPF events.**

| # | Particle source | Particle type | Carrier gas | Neutralizer | HR-DMA sheath flow (L min$^{-1}$) | HR-DMA aerosol flow (L min$^{-1}$) | Particle size range (nm) | Aerosol electrometer |
|---|---|---|---|---|---|---|---|---|
| 1 | Wire generator | Tungsten particles | $N_2$ | √ | 300, 145, 100 | 10, 6 | 1.2-20.3 | √ |
| 2 | Wire generator | Nickel/chromium particles | Synthesized air | √ | 200 | 10 | 1.2-14.0 | √ |
| 3 | Organic particle generator | Alpha-pinene oxidation particles | Synthesized air | √ | 150 | 5.4 | 1.6-9.4 | √ |
| 4 | Helsinki ambient air | Ambient particles | Atmospheric air | √ | 60 | 5 | 2-11 | |
| 5 | Hyytiälä ambient air | Ambient particles | Atmospheric air | √ | 60 | 5 | 2.5-11 | |

## 2.3 Campaign measurement

After the calibration, PSM 2.0 was operated in parallel with a DMPS to measure ambient particles in Hyytiälä (Fig. 1(b)). This lasted for three weeks from 1 May to 21 May, 2024. The instrument configurations are displayed in Table 2. Both the PSM 2.0 and DMPS used core-sampling method, with a carrier gas flow rate of 10.0 L min$^{-1}$ in the main sampling tube. PSM 2.0 was operated under the scanning mode, with the DEG saturator flow rate increasing from 0.05 to 1.80 L min$^{-1}$, and then decreasing. Each DEG scan took 4 minutes. A step inversion method was used for PSM 2.0 data inversion, based on calibration files using different types of particles.

**Table 2. Configuration settings for PSM 2.0 and DMPS during the ambient particle measurement in Hyytiälä.**

| Instrument | Core sampling | Neutralizer | Mode | Setting | Time per scan | Size bins |
|---|---|---|---|---|---|---|

| | | | | | | |
|---|---|---|---|---|---|---|
| PSM 2.0 | √ | | DEG scanning from 0.05 to 1.80 L min$^{-1}$ | Inlet/saturator/condenser = 40/71/10 (°C) * | 240 s | 6-9 |
| DMPS | √ | √ | Voltage scanning from 100 to 2000 V | Sheath/aerosol = 60:5(L min$^{-1}$) | 220 s | 10 |

* Under the boosted PSM temperature setting, the temperature of the condenser was 7 °C.

For DMPS (Kangasluoma et al., 2018), after the core-sampling, aerosol flow passed through a Ni[63] neutralizer at a flow rate of 5.0 L min$^{-1}$. Particles were then size-classified by a Half-mini DMA (sheath flow rate of 60 L min$^{-1}$). PSM 1.0 was used as the concentration detector of DMPS, which operated at a fixed DEG saturator flow rate of 0.3 L min$^{-1}$ (with a background concentration of almost 0 cm$^{-3}$). We used the DMPS inversion method as displayed in a previous study (Jiang et al., 2011). An equivalent length of 1.8 m was used to correct the diffusion losses inside the neutralizer. A charging steady state was assumed to be achieved inside the neutralizer (Wiedensohler and Fissan, 1991). The transmission function of Half-mini DMA was obtained from a previous study (Cai et al., 2018b). The detection efficiencies of different sized particles by the detector (PSM 1.0) were calibrated using alpha-pinene oxidation particles.

## 2.4 Data processing

### 2.4.1 Detection efficiency curve

For particles of a certain size, their detection efficiencies ($\eta$) at different DEG saturator flow rates were calculated based on the ratios of the concentrations measured by PSM 2.0 and actual concentrations. For metal particles and organic particles, the actual concentrations were typically higher than 1000 cm$^{-3}$, and could be measured by the AEM. However, in terms of ambient particles after DMA classification, their concentrations were identified based on the assumption that a stable plateau concentration observed under high DEG saturator flow rates represented the real particle concentrations. The detection efficiencies under different DEG saturator flow rates were fitted using the following Eq. (1):

$$\eta = \frac{1}{\left(1+e^{(-a\cdot(s-b))}\right)} ,$$

(1)

where $s$ is the DEG saturator flow rate, $a$ and $b$ are fitting parameters.

### 2.4.2 Kernel function curve

The Kernel function represents the derivative of the fitted detection efficiency curve with respect to the saturator flow rate (Cai et al., 2018b). The peak point of the Kernel function helps to establish the correlation between the DEG saturator flow rates and the corresponding cut-off sizes. Additionally, the width of the Kernel function suggests if the activation of particles happened within a narrow DEG saturator flowrate variation or a wide range. The sizing resolution in DEG saturator flow rate space ($Res(S^*)$) of PSM 2.0 can be evaluated, accordingly:

$$Res(S^*) = \frac{S^*}{\Delta s}$$

(2)

where $S^*$ is the saturator flow rate corresponding to the peak point of the Kernel function, and $\Delta S$ is the full width at half maximum of the Kernel function peak.

Several factors influence $Res(S^*)$, including the temperature configuration of the PSM, the uniformity and stability of DEG supersaturation within the instrument, and the uniformity of the size and chemical composition of the particles (Fernández de la Mora et al., 2022). In general, improving the uniformity and stability of DEG supersaturation within the PSM would enhance detection efficiency curve's sharpness, thereby enhancing the $Res(S^*)$. Conversely, variations in particle composition and poor monodispersity of particles can broaden the measured kernel function peaks, leading to a decrease in $Res(S^*)$. This study did not attempt to quantify the impact of these factors. Instead, we only characterized PSM 2.0's $Res(S^*)$ using metal particles under the standard temperature setting.

The $Res(S^*)$ alone cannot demonstrate the sizing capability of the PSM 2.0, because the relationship between particle sizes and DEG saturator flow rates is not linear. In terms of metal particles, slight variations in the saturator flow rate would lead to significant fluctuations in the corresponding cut-off sizes (Fig. 5). To address this, we cited the DMA's definition of size resolution, by replacing the DEG saturator flow rate in the Kernel function with the corresponding cut-off size (based on the calibration curve). After that, we calculated the size resolution of PSM ($Res(d_p^*)$) at specific particle size ($d_p^*$):

$$Res(d_p^*) = {d_p^*}\big/{\Delta d_p} = Res(S^*)\frac{d_p^*}{S^*}\frac{1}{-f'[S^*]}, \tag{3}$$

where $f[S^*]$ is the fitted function displaying the cut-off size at the saturator flow rate of $S^*$. $f'[S^*]$ suggests the derivation of the fitted function at the DEG saturator flow rate of $S^*$. The size resolution of PSM is related to the $Res(S^*)$, and also influenced by the calibration curve. The physical meaning of $Res(d_p^*)$ is the size range corresponding to the particle activation at a given DEG saturator flow rate. Higher size resolution means that each DEG saturator flow rate corresponds to a narrower size range near the cut-off size, and vice versa. Please refer to SI for the detailed derivation process.

### 2.4.3 Calibration curve

The calibration curve was based on the peak points of different-sized particles in the Kernel functions, and showed a one-to-one correspondence between the DEG saturator flow rate and cut-off size. Various types of particles were used to calibrate the PSM 2.0, resulting in different calibration curves. This calibration curve is crucial for PSM 2.0 data inversion. In terms of the step inversion method, the size resolution of the PSM 2.0 at each DEG saturator flow rate is assumed to be infinite. The activation of particles larger than the cut-off size are assumed to be 100%, while for particles smaller than the cut-off size, their activation efficiencies are assumed to be 0%.

# 3 Results and discussion

## 3.1 PSM 2.0 Calibration using metal particles

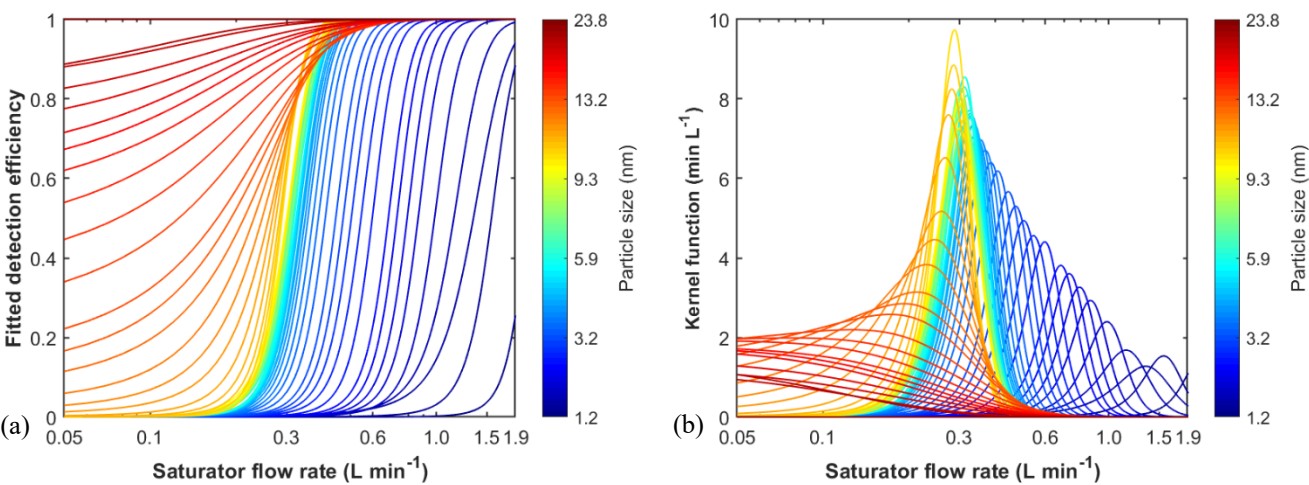

**Figure 2. (a) The fitted detection efficiency curves according to the calibration using tungsten particles. (b) The calculated Kernel function curves according to the fitted detection efficiencies curves.**

Figure 2(a) presents the detection efficiency curves for different sized tungsten particles. The detection efficiency increasing from 0% to plateau values (close to 100%) can be found as the DEG saturator flow rate increasing. The curves can be divided into two groups. The first group is for sub-10 nm particles, where the detection efficiency curves run approximately parallel to each other. This parallel pattern suggests that particle size is a key factor influencing activation. As particle size increase, a leftward shift in the detection efficiency curve is observed.

The second group is for particles larger than 10 nm. The curves deviate from the parallel pattern and begin to flatten as size increases. A plausible explanation is that the activation of particles above 10 nm is not solely determined by the DEG saturator flow rate but also by the downstream CPC. Particles larger than 10 nm start to be activated by the CPC with finite and increasing detection efficiencies. It hinders the establishment of a one-to-one relationship between each DEG saturator flow rate and its corresponding cut-off size.

In the calibration by using particles larger than 10 nm, we found the concentrations measured by the AEM started exceed those of PSM 2.0, with the difference increasing to 30% as particle size approached 20 nm. This discrepancy likely arises from the presence of multiply charged particles after the DMA classification, which can lead to an overestimation in concentrations by AEM. Consequently, in this size range, the concentrations measured by PSM 2.0 at the high DEG flow rates were adopted as the actual particle concentrations and used to plot the detection efficiency curves in Fig. 2(a).

$Res(S^*)$ of PSM 2.0 was calculated based on the Kernel function as displayed in Fig. 2(b). Notably, the $Res(S^*)$ remained stable for particle between 2.0 and 10.0 nm, approximately at 3 (Fig. 3). The decreasing trend in the $Res(S^*)$ for sub-2 nm particles may come from the changes in chemical compositions within this size range. To get a high concentration of sub-2

nm particles for instrument calibration, a higher heating power was used by the wire generator, which could increase the fraction of organic components in sub-2 nm metal particles. When sub-2 nm particles contain a mixture of metal and organic components, their activation corresponds to a broader range of DEG saturator flow rates. This leads to a flattening of the detection efficiency curve and a reduction in measured $Res(S^*)$. For particles larger than 10 nm, we also observed a decrease in the $Res(S^*)$. As discussed above, the activation of 10-20 nm particles is commonly influenced by the DEG saturator flow rate and the downstream CPC, leading to a flattening of the detection efficiency curves and a reduction in the $Res(S^*)$.

For sub-3 nm particles, the $Res(S^*)$ of PSM 2.0 were quite comparable with PSM 1.0 (Cai et al., 2018b). In 3 to 4 nm size range, a decrease in the $Res(S^*)$ can be found for PSM 1.0, whereas for PSM 2.0, the $Res(S^*)$ remained stable till 10 nm. PSM 2.0 showed a higher $Res(S^*)$ than PSM 1.0 possibly because PSM 2.0 ensures a more stable flow field and enables more precise and uniform control over DEG supersaturation. The PSM 2.0 can reach a much lower DEG supersaturation compared to the PSM 1.0, making it more suitable for measuring particles in the 3 to 10 nm range.

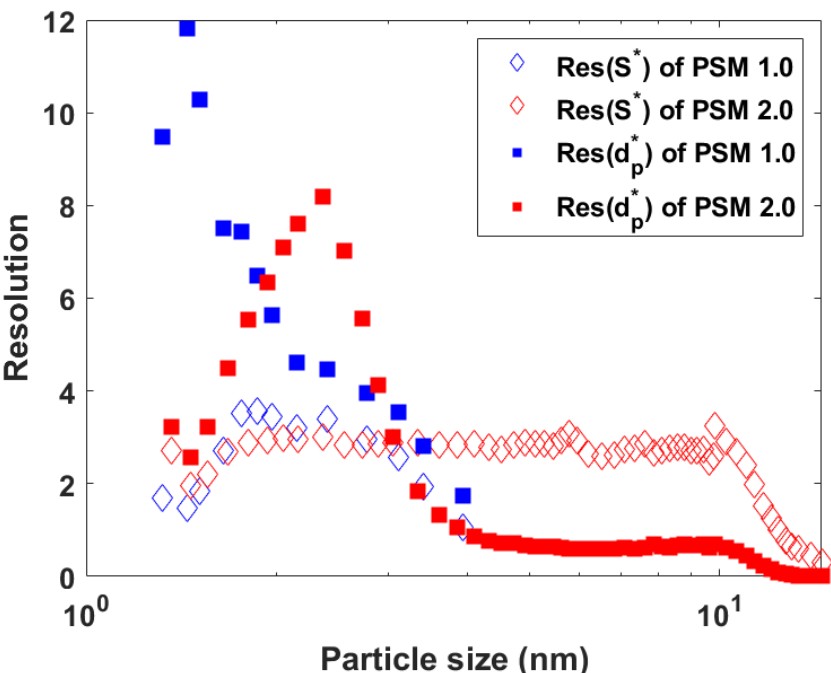

**Figure 3. Resolution in DEG saturator flow rate ($Res(S^*)$) and size resolution ($Res(d_p^*)$) for both PSM 2.0 and PSM 1.0. The calculations are based on metal particles classified by an HR-DMA. Commercial temperature settings were used for both instruments. The $Res(S^*)$ of PSM 1.0 are cited from (Cai et al., 2018a).**

The size resolution ($Res(d_p^*)$) of PSM 2.0 is a key factor in demonstrating its sizing ability for different particle sizes. Here, we compared the size resolution of PSM 2.0 with that of PSM 1.0. Since the working principle of the PSM is based on particle-vapor interaction under varying DEG supersaturation, this process is strongly influenced by the composition of the seed particles.

In general, higher $Res(d_p^*)$ were observed for small particles for bothe PSM 1.0 and 2.0. For PSM 1.0, a decreasing trend in size resolution was observed as particle size increased. For PSM 2.0, the peak size resolution was observed at around 2.2 nm.

The paek resolution was higher than 8, which is comparable or even higher than some types of DMA's resolution in this size (Cai et al., 2018a). The decrease in $Res(d_p^*)$ for particles smaller than 2.2 nm is similar to the decrease in $Res(S^*)$, as metal particles produced in the wire generator were contaminated by more organic impurities. For particles larger than 4 nm, the size resolution of PSM 2.0 was stable but was lower than 1. Though PSM 2.0 expanded the size measurement range from 4 nm of PSM 1.0 to above 10 nm, but the size resolution on 4-10 nm particles was low. This result provides useful insights into the

size bin selection for the PSM 2.0. For sub-3 nm particles, smaller size bins are recommended since PSM has higher size resolution in this range, while for particles in the 3 to 10 nm range, larger size bins are advisable.

For particles larger than 10 nm, the size resolution decreases further, resulting in larger sizing uncertainties. In summary, based on calibrations with various-sized metal particles, 10 nm is recommended as the upper size limit for PSM 2.0. This recommendation stems from the higher size resolution of PSM 2.0 in the sub-10 nm range and the fact that particles larger

than 10 nm begin to be activated by the CPC.

## 3.2 PSM 2.0 calibration with ambient particles

### 3.2.1 Ambient particles in Helsinki

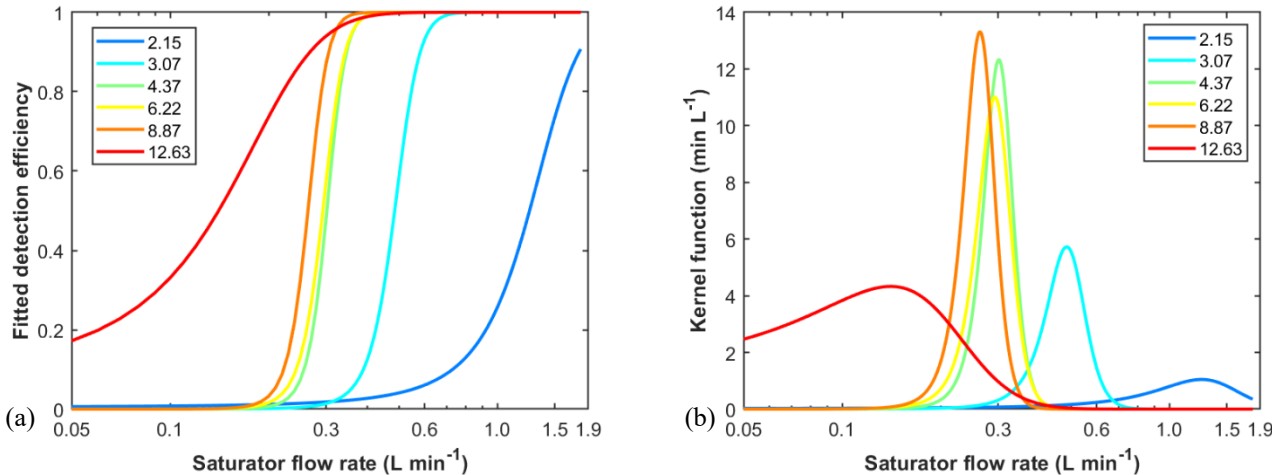

**Figure 4. PSM 2.0 calibration based on the ambient particles collected in Helsinki during an NPF event on 18 February. (a) Detection**
**efficiency curves of different sized particles. (b) Kernel function curves of different sized particles.**

Two NPF events in Helsinki were identified on 18 and 19 February, respectively. The size-resolved detection efficiency curves are displayed in Fig. 4 (a). Similar to the results of metal particle calibration, the detection efficiency curves of Helsinki ambient particles also moved toward left side, as particle size increasing. The corresponding Kernel function is plotted (Fig. 4 (b)). The main challenge of performing direct calibration using atmospheric particles is the low concentrations (Fig. S1). During NPF

events, the size-resolved concentrations of ambient particles were several magnitudes lower than those from the particle generator. After DMA classification, the concentrations of sub-2 nm particles approached 0 cm⁻³. When concentrations are very low, it becomes difficult to identify the actual values for calibration. Therefore, only particles larger than 2 nm were used for PSM 2.0 calibration.

We compared the calibration curves for all types of particles (Fig. 5). Overall, the calibration curves of metal particles, by

using tungsten and Ni/Cr particles, are comparable. Some differences are observed between metal particles and organic particles. The sub-4 nm organic particles needs higher DEG saturator flow rate compared to metal particles. This result is consistent with some previous studies that the activation of organic particles by PSM 1.0 will need higher DEG saturator flow rates (Kangasluoma et al., 2014). For particles larger than 4 nm, the calibration curve of organic particles is comparable or slightly lower than metal particles.

For atmospheric particles sized between 2-4 nm, the activation of Helsinki ambient particles is comparable with metal particles. The activation of particles larger than 4 nm will need lower DEG saturator flow rates, which could be related to the higher RH in the ambient atmosphere. The PSM detection efficiency increases with increasing relative humidity (Kangasluoma et al., 2013). Overall, the difference in the calibration curves between metal particles and Helsinki ambient particles is not substantial, which further confirmed the validity of using metal particles for PSM calibration, and subsequently using the PSM 2.0 to

measure atmospheric particles in Helsinki. However, the composition of ambient particles can vary significantly between different cities, as the sources of nucleation mode particles and the mechanisms of new particle formation (NPF) can differ. More direct calibration experiments using different urban atmospheric particles are strongly recommended.

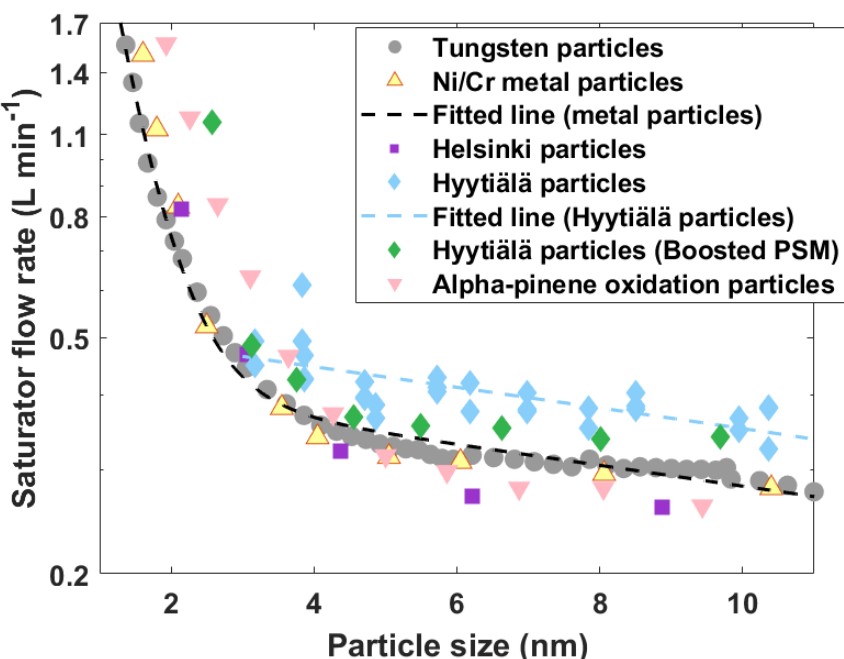

**Figure 5. The relationship between DEG saturator flow rates and the corresponding cut-off sizes. The calibrations curves were plotted based on different types of particles, including metal particles, organic particles, and ambient particles collected in different places.**

### 3.2.2 Ambient particles in Hyytiälä

Hyytiälä's atmospheric particles have also been used for PSM 2.0 calibration (Fig. S2-S3). A total of 6 NPF events were identified in Hyytiälä, on 5, 6, 10, 11, 12 March and 4 April, respectively. Among these, 5 events were used to calibrate PSM 2.0 under standard temperature settings, except the event on 4 April, that was used to calibrate the boosted PSM 2.0. The fitted detection efficiency curves and the raw data are given in the SI. Only particles larger than 3 nm were used for the calibration(Fig. 5), due to the low concentrations of sub-3 nm particles.

Notably, a higher saturator flow rate is needed for the activation of Hyytiälä ambient particles, compared with metal particles, organic particles, or the ambient particles in Helsinki. The ambient particles in Hyytiälä predominantly comprised of organic components, but the properties of organic particles in Hyytiälä  may different from the alpha-pinene oxidation particles generated in the lab. A plausible explanation is that organic particles formed through alpha-pinene oxidation were highly oxidized, resulting in activation behavior similar to that of metal particles. In contrast, atmospheric particles from Hyytiälä could had a lower oxidation state, and would require a higher DEG supersaturation for activation. This conclusion was further corroborated by the boosted PSM 2.0 experiment. By increasing the temperature difference between the saturator and condenser, the calibration curve of Hyytiälä ambient particles moves toward to the calibration curve of metal particles under standard temperature setting.

In summary, the calibration curves for different particle types show some variation. Metal particles, Helsinki ambient particles, and alpha-pinene oxidation particles larger than 4 nm display similar detection efficiency curves. However, Hyytiälä ambient particles and alpha-pinene oxidation particles smaller than 4 nm require higher DEG saturator flow rates for activation than metal particles. The composition of sub-10 nm particles as well as the corresponding properties will affect the calibration results of PSM 2.0.

### 3.3 Ambient particle size distribution measurement

During the field campaign, 4 NPF events in Hyytiälä were observed between 8 and 11 May. The results measured by PSM 2.0 were inverted using different calibration files. Subsequently, these inverted size distributions  by the PSM 2.0 were compared with the DMPS.

### 3.3.1 Total concentrations of 3-10 nm particles

Three calibration curves based on tungsten particles, Helsinki atmospheric particles, and Hyytiälä atmospheric particles were used for the data inversion.  Regarding the selection of 3 nm, PSM 2.0 clearly demonstrates the ability to measure particles as small as 1 nm, as shown in the metal particle results (Fig. 5). However, when performing direct calibration by using atmospheric particles, there is an insufficient signal intensity for sub-3 nm particles.

The lower size limits differ for different calibration files: for metal particles, the calibration is performed down to 1 nm, while for Helsinki and Hyytiälä particles, the smallest detectable sizes for calibration experiments were around 2 and 3 nm, respectively. To compare the total concentrations inverted from different calibration files, we selected a common measurement

size range of 3-10 nm (Fig. 6).

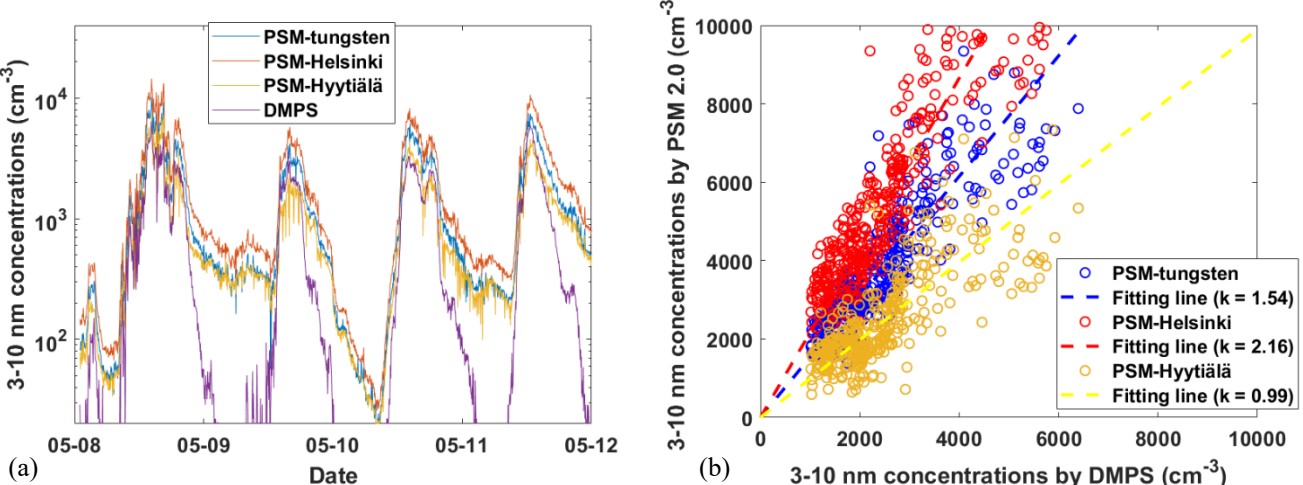

Figure 6. (a) 3-10 nm particle concentrations by PSM 2.0 and DMPS. Different types of particles were used for PSM 2.0 data inversion. (b) Comparison of total concentrations (sized between 3.1 to 10.1 nm) measured by DMPS and PSM 2.0 using different calibration methods. Only the DMPS total concentration results exceeding 1000 cm$^{-3}$ are shown. The fitting line is based on the

scattered data. A good correlation between PSM 2.0 and DMPS was observed, when Hyytiälä atmospheric particles was used for PSM 2.0 calibration.

During the NPF events, high concentrations of 3-10 nm particles were observed. The total concentrations of 3-10 nm particles are displayed in Fig. 6(a). The total concentrations by PSM and DMPS are scattered, when the total concentrations measured by DMPS were higher than 1000 cm$^{-3}$ (Fig. 6(b)). The best alignment between DMPS and PSM 2.0 measurements occurs

when using the Hyytiälä atmospheric particle calibration, showing a slope close to 1 in the linear regression analysis. In-situ calibration provides the most accurate total concentration measurements. In contrast, the calibration files of tungsten particle and Helsinki ambient particle result in overestimations by a factor of 1.54 and 2.16, respectively.

However, during non-NPF events, when total concentrations of sub-10 nm particles were lower than 1000 cm$^{-3}$, the concentrations measured by DMPS were significantly lower than those measured by PSM 2.0. In this condition, PSM 2.0 can

provide more reliable measurements than DMPS. This is because two instruments have different minimum concentration detection limit. The PSM can count single particles, resulting in a very low minimum concentration detection limit. Although sizing with PSM 2.0 is influenced by factors such as chemical composition, charging state, and relative humidity, the associated measurement uncertainties do not increase as concentrations decrease. PSM 2.0 can provide reliable total concentrations in both high and low ambient particle concentrations. In contrast, accurate measurements by DMPS require size-resolved particle

concentrations to be above the minimum detection limit (Kangasluoma and Kontkanen, 2017). For sub-10 nm particles, both

the charge fraction of nanoparticles in the neutralizer and the penetration efficiency through each component of the DMPS are low. The DMPS has a higher minimum concentration detection limit than PSM 2.0; if this limit is not met, its CPC may fail to detect any signal, leading to an underestimation of measured concentrations. This issue was particularly evident during our Hyytiälä campaign, where clean atmospheric conditions resulted in low sub-10 nm particle concentrations during non-NPF

events. Consequently, PSM is more suitable for measuring low concentrations of nanoparticles.

### 3.3.2 Particle number size distributions measured by PSM 2.0

The total concentrations were distributed into different size bins based on differernt calibration files. The inverted size distributions measured by PSM 2.0 and DMPS were displayed in Fig. 7. Significant differences were observed in the inverted particle size distributions of PSM 2.0 when using different calibration files. When the calibration files for tungsten particles or

Helsinki ambient particles were applied, the size-resolved concentrations showed an increasing trend as particle size decreased. In contrast, the opposite trend was observed when using the in-situ calibration file for Hyytiälä atmospheric particles. The validity of the in-situ calibration was confirmed through comparisons with DMPS measurements, which displayed a similar pattern in the 3-10 nm size range.

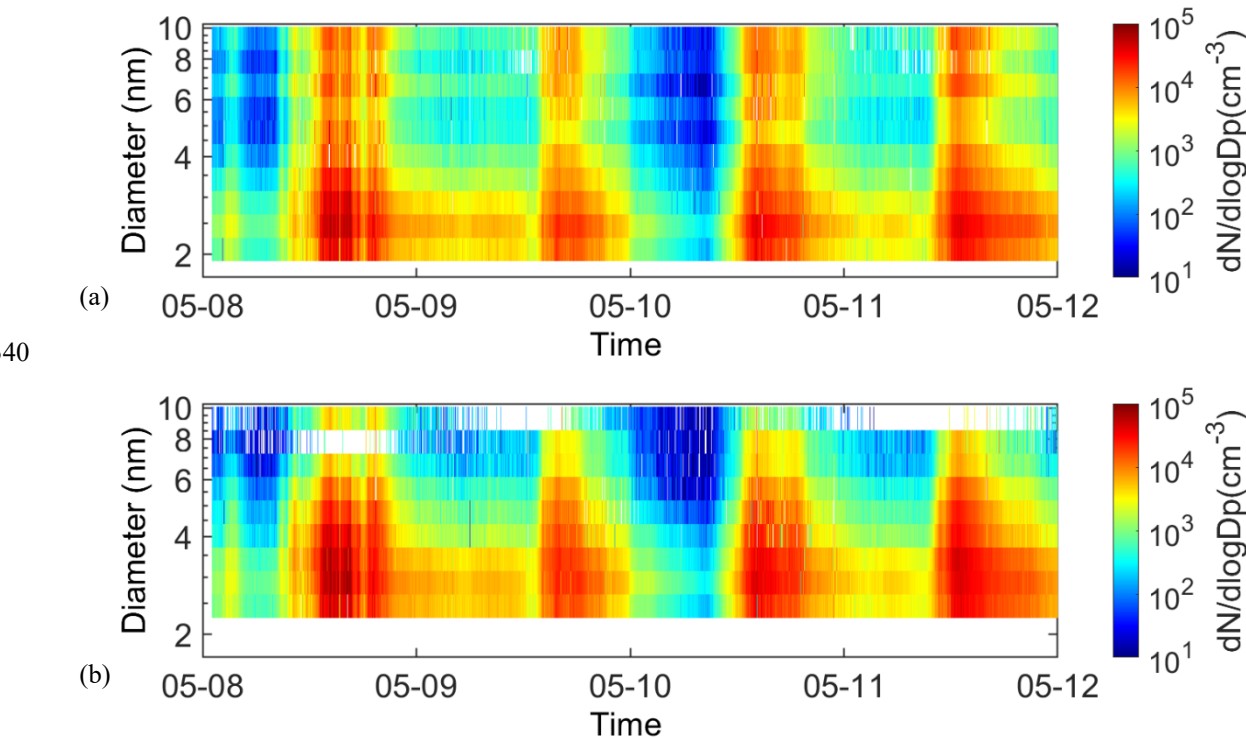

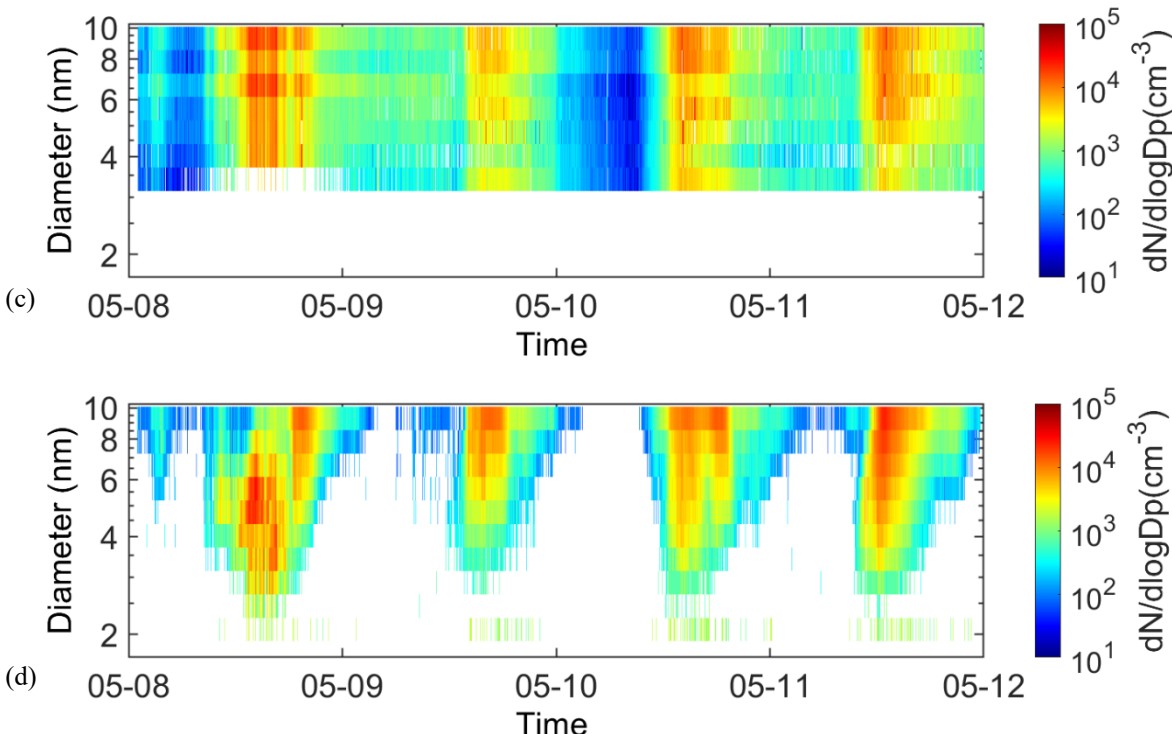

**Figure 7. (a) Number size distributions measured by PSM 2.0 using the calibration of tungsten particles. (b) Number size distributions measured by PSM 2.0 using the calibration of Helsinki ambient particles. (c) Number size distributions measured by PSM 2.0 using the calibration of Hyytiälä ambient particles. (d) Number size distributions measured by DMPS.**

To enhance the clarity of the intercomparison, we compared the mean number size distributions measured during four NPF events (from 13:00 to 15:00) (Fig. 8). Our result suggests that the inverted particle size distributions are sensitive to the

calibrations. In the sub-3 nm size range, PSM 2.0 measurements using the calibration for tungsten particles and Helsinki ambient particles exhibited higher concentrations compared to DMPS results. As displayed in Fig. 5, the Hyytiälä atmospheric particles would need a higher DEG saturator flow rate to be activated. By using the wrong calibration file, the ambient particles larger than 3 nm were wrongly attributed to sub-3 nm size range, which caused the overestimation of sub-3 nm particles.

In four NPF events, three events (from 9 to 11 May) exhibited a similar particle size distribution pattern between DMPS and

PSM 2.0 using the Hyytiälä atmospheric particle calibration, except for the event on 8 May. During that event, DMPS showed a peak concentration between 4-6 nm, with a noticeable decrease in concentration as particle size increased. Due to PSM 2.0's lower size resolution in this range compared to DMPS, the measured particle size distribution tended to flatten out.

We do not assert that the results displayed by DMPS are absolutely accurate, as DMPS itself is subject to inherent uncertainties. However, within the size range of 3-10 nm, DMPS (and SMPS) is the most widely used instrument, and its sizing is based on

electrical mobility classification, and theoritically accurate and reliable. Meanwhile, PSM 2.0 can provide comparable particle size distributions in this size range. Considering PSM 2.0 and DMPS are based on different working principles, this consistency

is noteworthy. This study offers a plausible explanation for the overestimation by PSM in the sub-4 nm size range in other studies, as well as the higher concentrations compared to DMPS.

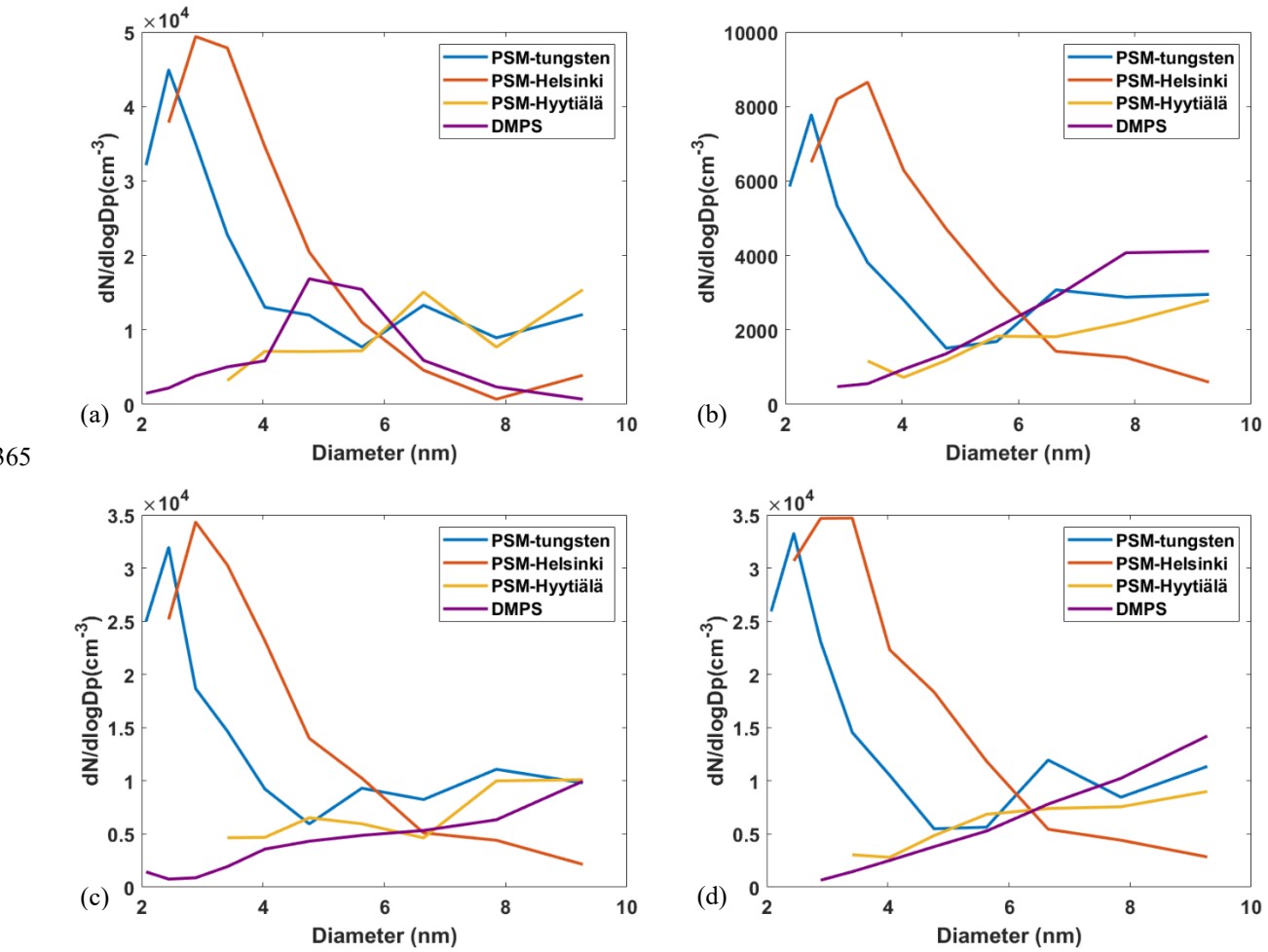

**Figure 8. Mean number size distributions measured by PSM 2.0 and DMPS. The calibration of PSM 2.0 was based on tungsten metal particles, Helsinki atmospheric particles, and Hyytiälä atmospheric particles, respectively. The size distributions were collected during NPF events that occurred (a) from 13:00 to 15:00 on 8 May, (b) from 13:00 to 15:00 on 9 May, (c) from 13:00 to 15:00 on 10 May, and (d) from 13:00 to 15:00 on 11 May.**

## 4. Conclusions

In this study, we calibrated the PSM 2.0 using different types of particles including tungsten particles, Ni/Cr particles, alpha-pinene oxidation particles, and atmospheric particles from Helsinki and Hyytiälä, respectively. Number size distributions of sub-10 nm particles based on different calibrations were investigated and compared with those measured by a DMPS. Calibration with Helsinki ambient particles showed a similar trend to metal particles, possibly because the composition of

urban particles has the similar activation behaviour with metal particles. However, the activation of Hyytiälä ambient particles required higher DEG saturator flow rates than other types of particles. This difference underscores the significance of particle composition in calibration processes. Proper calibration enables PSM 2.0 to enhance its reliability. After in-situ calibration (using the same atmospheric particles for calibration and subsequent measurements), PSM 2.0 exhibited a good correlation
with DMPS in terms of both total concentrations and particle size distributions, particularly during NPF events. However, using a wrong calibration will lead to deviations in both the inverted total concentrations and particle size distributions. PSM 2.0 also showed its effectiveness in measuring low concentrations of sub-10 nm particles. The lower size limit for direct calibration using atmospheric particles was between 2 and 3 nm, primarily due to observed NPF events not strong enough and a lack of particle concentration after DMA classification. More direct calibrations in differernt places around the world are
expected in the future to further reduce the measurement uncertainty of PSM 2.0.

**Code/Data availability**

The characterizations of the tested PSM 2.0 are shown in the figures. The codes for the inversion methods are available upon request.

**Author contribution**

JK and YL designed the experiments. YL, AY, FS, and LA carried them out. YL prepared the manuscript with contributions from all co-authors.

**Competing interests**

The authors declare that they have no conflict of interest.

**Disclaimer**

Publisher's note: Copernicus Publications remains neutral with regard to jurisdictional claims made in the text, published maps, institutional affiliations, or any other geographical representation in this paper. While Copernicus Publications makes every effort to include appropriate place names, the final responsibility lies with the authors

**Acknowledgements**

This work was funded by the Chongqing Natural Science Foundation (CSTB2022NSCQ-MSX1518) and National Natural
Science Foundation of China (42405112), and the Research Council of Finland (356134, 346370, 364223).

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
