# Peer review of "Direct calibration using atmospheric particles and performance evaluation of PSM 2.0 for sub-10 nm particle measurements"

_EGUsphere, 2024_

## Author Response (AR1)

RE: A point-to-point response to reviewers' comments

**Reviewer #1**

*The manuscript presents the calibration and application of the newly developed Particle Size Magnifier version 2.0 (PSM 2.0). The PSM 2.0 was calibrated by 1-10 nm particles, including metal and organic particles, and ambient particles collected from Helsinki and Hyytiälä. A Half-mini DMPS was also used to parallelly verify the PSM 2.0 in ambient observations. A key finding is that atmospheric particles from Hyytiälä required higher DEG (diethylene glycol) supersaturation for activation compared to standard metal particles, highlighting the impact of chemical composition of particles on measurement accuracy and emphasizing the need for in-situ calibration. The experiments are well made and analysis and interpretation appropriate. I would like to suggest a minor revision of the manuscript before the final acceptance.*

Reply: We thank the reviewer #1 for providing positive evaluation on our manuscript, and have addressed his/her concerns to improve the quality of the manuscript.

1. *The study mentions that a high-resolution DMA (HR-DMA) was used for the PSM calibration. However, different sheath flow rate was used in the HR-DMA (Table 1). While this may be effective in raising concentration (as the authors mentioned in the manuscript), it simultaneously reduces the resolution of the DMA. The authors should add some discussion on whether the change in the DMA resolution could affect the calibration results.*

Reply: We appreciate the reviewer's comment. In this study, different sheath and aerosol flow rates were applied based on the particle sizes used for calibration and their concentrations prior to DMA classification. Employing a large sheath/aerosol flow rate ratio (300/10 L min$^{-1}$/L min$^{-1}$) was essential for classifying sub-3 nm particles, ensuring that particles with good monodispersity used for calibrating the PSM 2.0. However, for particles larger than 10 nm, the voltage required for classification at a sheath flow rate of 300 L min$^{-1}$ exceeds the voltage limits of the DMA's power supply or the breakdown voltage between the DMA electrodes. Therefore, a decrease in sheath flow rate was utilized for particle size classification as particle size increased. For direct calibration using atmospheric particles, enhancing penetration efficiency through the HR-DMA is crucial to obtain a sufficient signal for calibration. A sheath-to-aerosol flow rate ratio of 60/5 (L min$^{-1}$/L min$^{-1}$) is recommended, providing reasonable size resolution and particle concentration for PSM 2.0 calibration.

We have revised our manuscript which reads (Line 105-107): "A decreased DMA sheath/aerosol flow rate ratio of 60/5 L min$^{-1}$/ L min$^{-1}$ was used for the classification of atmospheric particles, which would help to increase the penetration efficiency of nanoparticles passed through the DMA, and obtain a sufficient concentration for calibration."

2. *(Section 3.3) The manuscript presents a comparison between PSM and Half-mini DMPS, particularly during new particle formation (NPF) events. However, the comparison during non-NPF*

*events, when nanoparticle concentrations are lower, remains unclear. The authors should clarify which instrument is more reliable under such conditions.*

Reply: We appreciate the reviewer's insightful comment. During new particle formation (NPF) events, when sub-10 nm particles were present in high concentrations, both the PSM and Half-mini DMPS exhibited similar trends in total particle concentrations. However, a notable discrepancy arose during non-NPF events, with PSM reporting higher concentrations compared to DMPS. The PSM results were more accurate when ambient particle concentrations were low.

We have revised our manuscript accordingly, and added some discussion, which reads (Line 350-363): "However, during non-NPF events, when total concentrations of sub-10 nm particles were lower than 1000 cm$^{-3}$, the concentrations measured by DMPS were significantly lower than those measured by PSM 2.0. In this condition, PSM 2.0 can provide more reliable measurements than DMPS. This is because two instruments have different minimum concentration detection limit. The PSM can count single particles, resulting in a very low minimum concentration detection limit. Although sizing with PSM 2.0 is influenced by factors such as chemical composition, charging state, and relative humidity, the associated measurement uncertainties do not increase as concentrations decrease. PSM 2.0 can provide reliable total concentrations in both high and low ambient particle concentrations. In contrast, accurate measurements by DMPS require size-resolved particle concentrations to be above the minimum detection limit (Kangasluoma and Kontkanen, 2017). For sub-10 nm particles, both the charge fraction of nanoparticles in the neutralizer and the penetration efficiency through each component of the DMPS are low. The DMPS has a higher minimum concentration detection limit than PSM 2.0; if this limit is not met, its CPC may fail to detect any signal, leading to an underestimation of measured concentrations. This issue was particularly evident during our Hyytiälä campaign, where clean atmospheric conditions resulted in low sub-10 nm particle concentrations during non-NPF events. Consequently, PSM is more suitable for measuring low concentrations of nanoparticles."

3.  *(Line 85) Metal particles are commonly used for the calibration of PSM 2.0 and other particle counters. Could the authors provide an explanation as to why metal particles are preferred. It would greatly enhance the understanding of instrument calibration.*

Reply: Metal particles are commonly used for instrument calibration due to several key advantages over other particle types. Firstly, the wire generator (or an oven generator) can produce particles covering the size range for PSM calibration, from 1 nm to several tens of nanometers. One can select specific sizes using a HR-DMA for instrument calibration. Secondly, metal particle concentrations are stable and adjustable, easily modified by controlling the wire heating power and carrier gas flow rate. Lastly, studies have shown that the activation properties of metal particles are comparable to those of salt particles, which is assumed to be the main composition of ambient particles. However, the ambient particle compositions varied a lot in different places. So, comparing the calibration by using metal particles and direct calibration by using ambient particles is quite meaningful.

We have revised our manuscript accordingly, which reads (Line 94-98): "The compositional differences between laboratory-generated particles and atmospheric particles have been the main source of uncertainty in PSM measurements. Ambient particle compositions vary significantly across different locations, highlighting the value of performing direct calibration using atmospheric particles. A burst increase in sub-10 nm particle concentrations was observed during NPF events in Helsinki and Hyytiälä, with atmospheric particles being sampled directly for the calibration of PSM 2.0."

4. *(Line 96) The temperature setting of PSM 2.0 is of great interest to many users, as it is crucial to the instrument's performance. The authors should provide a more detailed explanation or reference to the standard temperature settings.*

Reply: The standard temperature setting used in this study was provided by Airmodus Inc. It was determined based on two main factors. First, PSM 2.0 operates at a higher condenser temperature (10°C) than PSM 1.0 (5°C) to mitigate the effects of co-condensation between diethylene glycol (DEG) and water molecules. Second, the saturator temperature was carefully adjusted to enable the variation of DEG saturator flow rate would cover the activation of 1 nm particles and particles larger than 10 nm.

The boosted temperature setting decreased the condenser temperature to threshold of DEG homogeneous nucleation (7°C), enhancing detection efficiency for particles near 1 nm. However, this adjustment also reduces the upper size limit of the PSM 2.0.

We have revised our manuscript, which reads (Line 120-127): "The standard temperature settings by the manufacturer were used for all calibration experiments, and one more calibration was performed under boosted temperature setting in Hyytiälä. The temperatures for both the standard and boosted settings are displayed in Table 2. The standard temperature setting established a condenser temperature of 10°C to reduce the potential co-condensation between diethylene glycol (DEG) and water molecules at peak DEG saturator flow rates. The saturator temperature was carefully adjusted to ensure that the scanning DEG saturator flow rate (from 0.05 to 1.90 L min$^{-1}$) could activate particles both near 1 nm and larger than 10 nm. The boosted temperature setting approached the threshold for DEG homogeneous nucleation at peak DEG saturator flow rates, enhancing PSM's detection efficiency for particles closing to 1 nm, meanwhile reducing the upper size limit."

**Reviewer #2**

*This is a most relevant article where a mixing CPC (PSM2) capable of sizing particles is calibrated with a diversity of aerosols, and then used to study atmospheric particles in conjunction with a better-established mobility-based sizing instrument (DMPS). The fact that calibrations have been obtained with several kinds of ambient particles is noteworthy, particularly in the case of ambient particles freshly formed in a boreal forest. This splendid feat was achieved by taking advantage of several new particle formation (NPF) events. A substantial discrepancy between the activation characteristics of these pristine particles and other aerosol sources is discovered, and the sizing difficulties associated to this strong material dependence is discussed. This finding poses a major challenge to current efforts at developing reliable methods to size particles. This I do not view at all as a weakness of the article, as it may provide the needed stimulus to identify alternative working fluids, hopefully less dependent on the aerosol composition.*

Reply: We thank Reviewer #2 for the positive evaluation of our manuscript, and have made revisions to improve the overall quality.

Regarding alternative working fluids for the PSM, we conducted tests using propylene glycol. We compared the uncertainties in sub-10 nm particle measurements with different working fluids. Although these results are not included in this paper, they will be published in the future.

*In spite of its high interest, the article has two problems that must be solved before it is ready for publication.*

1. *The writing is not up to the standards of more than one experienced co-author of the paper. There are problems both with the English style as well as with the clarity and the logical order of the text. There are important findings insufficiently emphasized as well as repetitions of less important matters. These reveal that the experienced coauthors of the article have not sufficiently reviewed and improved their own work before presenting it openly for discussion. A more serious involvement of these coauthors in a major rewriting is badly required before the presentation of the article is up to the standards of the results presented.*

Reply: We have made comprehensive revisions throughout the manuscript, focusing on improving the logical structure and clarity of the text. Co-authors have been actively involved in this process to enhance the overall quality of the manuscript.

2. *A second problem is that the sizing range covered appears to extend into particles large enough to be counted by the CPC detector, even without the PSM. This is manifested in Figure 2a by a tail on the left that does not decay to zero. This simply means that arbitrarily large particles are detected with finite efficiency and interpreted as if they were smaller than 10 nm. I suppose the authors are aware of this problem, but their choice of 10 nm as the upper size range is made without sufficient discussion to persuade the reader that this is not a problem.*

*Also, the dependence of this upper range limit on aerosol composition is not discussed.*

*Figure 6 shows that the total concentration given by the PSM is comparable to and approximately proportional to that of the DMPS at the beginning of each NPF event, when the particles are probably small. Yet, the discrepancy is of orders of magnitude as the event matures, when the particles are expected to be large. How do the authors explain this large discrepancy if not as due to the large particle counting problem noted in (2)? It is notorious that this problem applies to all the calibration methods included.*

Reply: 10 nm is identified as the upper size limit for PSM 2.0. While the PSM can measure particles slightly larger than 10 nm, this comes with increased measurement uncertainties. The recommended size measurement range for PSM 2.0 is based on calibrations with metal particles ranging from 1.2 to 23.8 nm. The 10 nm upper limit is suggested due to the reasonable size resolution (Figure 3) and the fact that particles larger than 10 nm started to be activated by the downstream CPC. PSM 2.0 measurements exhibit higher uncertainties for 10-20 nm particles compared to sub-10 nm particles. As particle size increases from 10 to 20 nm, the size resolution of PSM 2.0 decreases significantly, introducing further uncertainties in the measurements. Identifying 10 nm as the upper range limit under standard temperature settings is a conservative estimate.

We have revised our manuscript accordingly, which reads (Line 206-210): "The second group is for particles larger than 10 nm. The curves deviate from the parallel pattern and begin to flatten as size increases. A plausible explanation is that the activation of particles above 10 nm is not solely determined by the DEG saturator flow rate but also by the downstream CPC. Particles larger than 10 nm start to be activated by the CPC with finite and increasing detection efficiencies. It hinders the establishment of a one-to-one relationship between each DEG saturator flow rate and its corresponding cut-off size."

And (Line 253-255): "This result provides useful insights into the size bin selection for the PSM 2.0. For sub-3 nm particles, smaller size bins are recommended since PSM has higher size resolution in this range, while for particles in the 3 to 10 nm range, larger size bins are advisable."

And (Line 255-259): "For particles larger than 10 nm, the size resolution decreases further, resulting in greater sizing uncertainties. In summary, based on calibrations with various-sized metal particles, 10 nm is recommended as the upper size limit for PSM 2.0. This recommendation stems from the higher size resolution of PSM 2.0 in the sub-10 nm range and the fact that particles larger than 10 nm begin to be activated by the CPC."

And (Line 316-321): "In summary, the calibration curves for different particle types show some variation. Metal particles, Helsinki ambient particles, and alpha-pinene oxidation particles larger than 4 nm display similar detection efficiency curves. However, Hyytiälä ambient particles and alpha-pinene oxidation particles smaller than 4 nm require higher DEG saturator flow rates for activation than metal particles. The composition of sub-10 nm particles as well as the corresponding properties will affect the calibration results of PSM 2.0. "

In general, the sizing of sub-10 nm particles is based on the concentration differences measured under different DEG saturator flow rates. Larger particles, such as 100 nm, are fully activated by the CPC and measured at the same concentrations across different DEG saturator flow rates, meaning they are not assigned to any sub-10 nm size bin.

Based on the criteria used to identify the upper size limit of PSM 2.0, this limit is primarily determined by the cut-off size of the downstream CPC. The minimum particle size detectable by the CPC is comparable to the upper size limit of the PSM. The CPC utilizes butanol as the working fluid, and previous studies have shown that butanol-based CPCs are generally insensitive to particle composition (Liu et al., 2021), so the upper size limits should be comparable across different particle types. Our unpublished experiments suggest that reducing the temperature difference between the CPC's saturator and condenser shifts the detection efficiency curve of CPC toward larger particle sizes, extending the PSM's upper range limit by several nanometers. The effects of temperature setting on the performance of PSM 2.0 was not discussed in detail in this study.

For concentration discrepancies between PSM 2.0 and DMPS, please see our response to Reviewer #1's comment 2. Discrepancies within PSM 2.0, using different calibration files, result from the relationship between DEG saturator flow rates and corresponding cut-off sizes. Concentrations based on in-situ calibrations are considered more reliable. Please read (Line 374-378): "Significant differences were observed in the inverted particle size distributions of PSM 2.0 when using different calibration files. When the calibration files for tungsten particles or Helsinki ambient particles were applied, the size-resolved concentrations showed an increasing trend as particle size decreased. In contrast, the opposite trend was observed when using the in-situ calibration file for Hyytiälä atmospheric particles. The validity of the in-situ calibration was confirmed through comparisons with DMPS measurements, which displayed a similar pattern in the 3-10 nm size range."

3. *The fact that the article includes members of the company manufacturing the PSM, and that no conflict of interest is reported, suggests that special care should be exerted to avoid claiming that the sizing range of this instrument exceeds what it can credibly achieve in practice.*

Reply: We thank the reviewer's comment. In this study, the technicians at Airmodus provided the standard temperature settings for PSM 2.0. The 10 nm upper size limit for PSM 2.0 was conservatively determined based on the calibration results, without exaggeration of the instrument's performance.

A few minor remarks follow:

4. *"fitted detection efficiency curves maintain a consistent and approximate parallel pattern." The meaning of this parallel pattern is insufficiently clear.*

Reply: Thank you for the comment. The "parallel pattern" refers to the consistent shape of the fitted detection efficiency curves across different particle sizes, where the trends remain similar but shift

proportionally as particle size changes. We have revised the manuscript to make this explanation clearer, which reads (Line 198-202): "Figure 2(a) presents the detection efficiency curves for different sized tungsten particles. The detection efficiency increasing from 0% to plateau values (close to 100%) can be found as the DEG saturator flow rate increasing. The curves can be divided into two groups. The first group is for sub-10 nm particles, where the detection efficiency curves run approximately parallel to each other. This parallel pattern suggests that particle size is a key factor influencing activation."

5. *"Due to variations in detection efficiency curves for particles of varying sizes, PSM 2.0 can theoretically detect particles ranging from 1 to 20 nm. However, calibrating particles within the 10-20 nm range yields higher uncertainties. Concentrations measured by the AEM were gradually higher than PSM 2.0, as particle size closing to 20 nm. This discrepancy probably arises from the multiply charged particles, leading to overestimation in concentrations by AEM. Consequently, the concentrations measured by PSM 2.0 at the high DEG flow rates were adopted as the actual particle concentrations and used to plot the detection efficiency curves."*

   *The idea that the sizing range is extendible to 20 nm does not fit well with my discussion of (2). The notion that this simply increases uncertainty is incompatible with the fact that 100 and 200 nm particles will be counted as if they were 15 nm. The meaning of the rest of the paragraph is unclear to me.*

Reply: Please refer to our response to your comment 2. In this study, 10 nm was identified as the upper range limit because the sizing of sub-10 nm particles was not influenced by the CPC. For particles larger than 20 nm, the detection efficiency curves were nearly flat, indicating that PSM 2.0 lost its sizing ability for these larger particles. For particles sized between 10 and 20 nm, some differences in the detection efficiency curves can be observed, suggesting that PSM 2.0 retains some sizing ability for these particles, albeit with increased uncertainties (refer to the discussion on the size resolution of PSM 2.0). Consequently, we do not recommend using PSM 2.0 for particles larger than 10 nm.

For metal particle calibration, actual concentrations were measured using the AEM. However, we found that AEM concentrations could be overestimated for particles sized at 20 nm, as some particles may carry more than one charge. Therefore, we used the concentrations measured by PSM 2.0 under the high DEG saturator flow rates as the actual particle concentration for the detection curve plot in Figure 2.

We have revised our manuscript, which reads (Line 216-221): "In the calibration by using particles larger than 10 nm, we found the concentrations measured by the AEM started exceed those of PSM 2.0, with the difference increasing to 30% as particle size approached 20 nm. This discrepancy likely arises from the presence of multiply charged particles after the DMA classification, which can lead to an overestimation in concentrations by AEM. Consequently, in this size range, the concentrations measured by PSM 2.0 at the high DEG flow rates were adopted as the actual particle concentrations and used to plot the detection efficiency curves in Fig. 2(a)."

6. *"for particles larger than 3 nm, the detection efficiency curves can be well fitted, because a plateau*

*value in the detection efficiency curve can be well identified. However, for sub-3 nm particles, especially for sub-2 nm particles, the particles' concentrations were low."*

*I do not see what can be the connection between what is said before and after the "However".*

Reply: We appreciate the reviewer's comments. The actual particle concentrations are important for the plot of calibration curves, in the absence of AEM. For the ambient particles, the actual particle concentrations are identified based on the shape of the concentrations curve under different DEG saturator flow rates. If the concentrations measured by the PSM increased gradually and reach the plateau values, the plateau values will be identified as the actual particle concentrations.

The lack of signal intensity is the main challenge for direct calibration. When concentrations after DMA classification approach 0 cm⁻³, measurements are influenced by random counting, making it impossible to identify the plateau value or determine the actual concentrations. For the direct calibration using Helsinki atmospheric particles, the lower size limit was around 2 nm, while for Hyytiälä ambient particles, it was 3 nm.

We have revised our manuscript accordingly (Line 267-273): "The main challenge of performing direct calibration using atmospheric particles is the low concentrations. During NPF events, the size-resolved concentrations of ambient particles were several magnitudes lower than those from the particle generator. After DMA classification, the concentrations of sub-2 nm particles approached 0 cm⁻³. When concentrations are very low, it becomes difficult to identify the actual values for calibration. Therefore, only particles larger than 2 nm were used for PSM 2.0 calibration."

7. *"By increasing the temperature difference between the saturator and condenser, the calibration curve moves toward to the calibration curve of metal particles."*
   *To mean something concrete this sentence needs several clarifications. Which calibration curve? Hyytiälä particles or laboratory alpha-pinene oxidation particles? And the calibration of metal particles referred to, is it with or without increasing the temperature difference?*

Reply: We apologize for the lack of clarity in our previous statement. Please read (Line 310-315): "A plausible explanation is that organic particles formed through alpha-pinene oxidation were highly oxidized, resulting in activation behavior similar to that of metal particles. In contrast, atmospheric particles from Hyytiälä could had a lower oxidation state, and would require a higher DEG supersaturation for activation. This conclusion was further corroborated by the boosted PSM 2.0 experiment. By increasing the temperature difference between the saturator and condenser, the calibration curve of Hyytiälä ambient particles moves toward to the calibration curve of metal particles under standard temperature setting."

**References**

Kangasluoma, J. and Kontkanen, J.: On the sources of uncertainty in the sub-3nm particle concentration measurement, J. Aerosol Sci., 112, 34–51, https://doi.org/10.1016/j.jaerosci.2017.07.002, 2017.

Liu, Y., Attoui, M., Li, Y., Chen, J., Li, Q., and Wang, L.: Characterization of a Kanomax® fast condensation particle counter in the sub-10 nm range, J. Aerosol Sci., 155, 105772, https://doi.org/10.1016/j.jaerosci.2021.105772, 2021.